

# Effects of extreme meteorological factors and high air pollutant concentrations on the incidence of hand, foot and mouth disease in Jining, China

Haoyue Cao[1,*], Rongrong Xu[2,*], Yongmei Liang[3], Qinglin Li[1], Wenguo Jiang[4], Yudi Jin[5], Wenjun Wang[6] and Juxiang Yuan[1]

[1] School of Public Health, North China University of Science and Technology, Tangshan, Hebei Province, China
[2] State Key Laboratory of Environmental Criteria and Risk Assessment, Chinese Research Academy of Environmental Sciences, Beijing, China
[3] Business Management Department, Jining Center For Disease Control And Prevention, Jining, Shandong, China
[4] Infectious Disease Prevention and Control Department, Jining Center For Disease Control And Prevention, Jining, Shandong, China
[5] Department of Radiology, The First Affiliated Hospital of Chongqing Medical University, Chongqing, China
[6] Weifang Nursing Vocational College, Weifang, Shandong, China
* These authors contributed equally to this work.

Corresponding authors
Wenjun Wang,
wwjun1973@163.com
Juxiang Yuan, yuanjx@ncst.edu.cn

## ABSTRACT

**Background:** The evidence on the effects of extreme meteorological conditions and high air pollution levels on incidence of hand, foot and mouth disease (HFMD) is limited. Moreover, results of the available studies are inconsistent. Further investigations are imperative to elucidate the specific issue.

**Methods:** Data on the daily cases of HFMD, meteorological factors and air pollution were obtained from 2017 to 2022 in Jining City. We employed distributed lag nonlinear model (DLNM) incorporated with Poisson regression to explore the impacts of extreme meteorological conditions and air pollution on HFMD incidence.

**Results:** We found that there were nonlinear relationships between temperature, wind speed, $PM_{2.5}$, $SO_2$, $O_3$ and HFMD. The cumulative risk of extreme high temperature was higher at the 95th percentile ($P_{95}$th) than at the 90th percentile ($P_{90}$th), and the $RR$ values for both reached their maximum at 10-day lag ($P_{95}$th $RR$ = 1.880 (1.261–2.804), $P_{90}$th $RR$ = 1.787 (1.244–2.569)), the hazardous effect of extreme low temperatures on HFMD is faster than that of extreme high temperatures. The cumulative effect of extreme low wind speeds reached its maximum at 14-day lag ($P_{95}$th $RR$ = 1.702 (1.389-2.085), $P_{90}$th $RR$ = 1.498 (1.283–1.750)). The cumulative effect of $PM_{2.5}$ concentration at the $P_{90}$th was largest at 14-day lag ($RR$ = 1.637 (1.069–2.506)), and the cumulative effect at the $P_{95}$th was largest at 10-day lag ($RR$ = 1.569 (1.021–2.411)). High $SO_2$ concentration at the $P_{95}$th at 14-day lag was associated with higher risk for HFMD ($RR$: 1.425 (1.001–2.030)).

**Conclusion:** Our findings suggest that high temperature, low wind speed, and high concentrations of PM2.5 and SO2 are associated with an increased risk of HFMD. This study not only adds insights to the understanding of the impact of extreme meteorological conditions and high levels of air pollutants on HFMD incidence but

also holds practical significance for the development and enhancement of an early warning system for HFMD.

# INTRODUCTION

Hand, foot, and mouth disease (HFMD) represents a highly contagious ailment primarily instigated by Coxsackievirus A16 (CV-A16) and Enterovirus 71 (EV-71) (*Kimmis, Downing & Tyring, 2018*). The primary mode of transmission is contact-based, with a predilection for affecting individuals under the age of five (*World Health Organization (WHO), 2018*). The majority of HFMD cases exhibit an incubation period ranging from 3 to 10 days, characterized by prominent symptoms such as fever and the development of vesicular lesions on the hands, feet, and oral mucosa (*Gu, Li & Lu, 2020*; *Cox & Levent, 2018*). While the overall prognosis for HFMD is generally favorable, it is noteworthy that the disease may elicit severe manifestations, including neurological symptoms, thereby imparting a substantial disease burden upon both individuals and society at large (*Ooi et al., 2010*; *Huang et al., 1999*).

In the past few decades, several epidemics and outbreaks of HFMD had occurred worldwide since the first report of HFMD in New Zealand in 1957 (*Zhuang et al., 2015*). Since the beginning of the 21st century, the proportion of severe cases of HFMD in the Asia-Pacific region has been unusually high and the epidemic has continued (*Yi et al., 2020*). The burden of HFMD in China is about 75,881 disability-adjusted life years per year, the largest proportion among many high-burden countries (*Koh et al., 2018*). From 2008 to the end of 2019, more than 22 million cases of HFMD were reported, resulting in 3,500 deaths. In 2021, it was listed as a category C infectious disease with the highest incidence in China (*The Chinese Bureau for Disease Prevention and Control, 2020*; *National Health Commission of the People's Republic of China, 2022*).

In numerous epidemiological investigations, the close association between the survival and transmission of the HFMD pathogen and the surrounding environment has been consistently demonstrated (*Tian et al., 2018*; *Zhong et al., 2018*). In recent years, there has been a heightened focus on the impact of meteorological factors on infectious diseases. Previous research has presented compelling evidence regarding the correlation between HFMD incidence and meteorological factors. For instance, a study conducted in Minhang District, Shanghai, identified a positive correlation between HFMD and wind speed, coupled with a negative correlation with precipitation and sunshine hours (*Qi et al., 2018*). However, research exploring the relationship between HFMD incidence and weather extremes remains limited. Extreme weather events, typically occurring less frequently than 90 or 10 percent of the probability density function estimated from observations (*Chen et al., 2020*), have not been extensively investigated in the context of HFMD. A multi-city study in mainland China unveiled a non-linear relationship between temperature and

HFMD, where high temperatures were recognized as a risk factor for the disease (*Xiao et al., 2017*). Low temperatures were found to exhibit a protective effect on HFMD in Shijiazhuang, while posing a risk in Wuhan (*Liu et al., 2022*; *Hao et al., 2020*). Furthermore, a study in Hefei identified low wind speed as a risk factor for HFMD development (*Zhang et al., 2019*), which contradicts the findings of a study in Guilin, where low wind speed was deemed a protective factor for HFMD development (*Yu et al., 2018*). Consequently, the results from various studies investigating the relationship between HFMD and meteorological factors exhibit significant heterogeneity, underscoring the imperative for additional research to comprehensively explore these intricate associations.

Compared to meteorological factors, the relationship between HFMD and air pollutants has been less well studied. Air pollution has been shown to have adverse effects on the respiratory and cardiovascular systems (*Chen et al., 2020*; *Cheng et al., 2018*). A study in Shanghai found that the higher the $PM_{2.5}$ concentration, the more years of human life lost (*Ruan et al., 2020*). Currently, there are few studies on air pollutants and HFMD and the findings vary widely. *Qian et al. (2023)* found that high concentrations of $PM_{2.5}$ can reduce the incidence of HFMD, but in *Peng et al. (2022)*, high concentrations of $PM_{2.5}$ was risk factor for the incidence of HFMD. In addition, studies of the relationship between other high-level pollutants and HFMD vary widely. For example, a study in Hefei (*Wei et al., 2019*) found that high concentrations of $SO_2$ increased the risk of HFMD, while a study in Shijiazhuang (*Liu et al., 2022*) reached the opposite conclusion. A study in Ningbo (*Huang et al., 2016*) found that high concentrations of $O_3$ increased the risk factors for HFMD incidence. However, a study in Guilin (*Yu et al., 2019*) obtained that high concentrations of $O_3$ decreased HFMD incidence. Therefore, more research is needed to elucidate the relationship between high concentrations of air pollutants and HFMD.

In this study, we analyzed the relationship between the incidence of HFMD and different extreme meteorological conditions as well as varying concentrations of pollutants. The intention is to furnish a foundation for the proactive implementation of warning systems to prevent the onset and progression of HFMD.

## MATERIALS AND METHODS

### Study area
Jining City is situated in the southern region of Shandong Province, China (34°25′~35°55′ N, 115°54′~117°06′ E), encompassing an area of 11,191 square kilometers and a population of 5.1 million. The administrative divisions include two districts (Rencheng and Yanzhou), two county-level cities (Qufu and Zoucheng), and seven counties (Sishui, Weishan, Yutai, Jinxiang, Jiaxiang, Wenshang, and Liangshan). All residents of the above areas were selected for this study. Characterized by a typical temperate monsoon climate, Jining experiences hot and rainy summers, as well as cold and dry winters, with an average annual temperature of approximately 13 °C (*Li et al., 2016*).

## Data collection

The HFMD case data spanning from 2017 to 2022 were sourced from the infectious disease surveillance system operated by the Jining Center for Disease Control and Prevention. In China, HFMD attained classification as a Class C infectious disease in 2008, with the surveillance data being consistently updated on a daily basis, ensuring high-quality data. The clinical diagnostic criteria for HFMD adhere to the Guidelines for the Prevention and Treatment of Hand, Foot, and Mouth Disease as prescribed by the Ministry of Health of China.

The corresponding meteorological data were from NASA (https://www.nasa.gov/). The air pollutant data were from the China Air Quality Online Monitoring and Analysis Platform Historical Data (https://www.aqistudy.cn/). The meteorological data include the mean temperature (°C) daily, barometric pressure (kPa), relative humidity (%), and wind speed (m/s), while the air pollutant data include daily concentration data of $PM_{2.5}$, $PM_{10}$, $NO_2$, $SO_2$, and average concentration of $O_3$ within 8 consecutive hours. These data were provided by the official website, so the data are authoritative, authentic and timely.

The meteorological data were procured from NASA (https://www.nasa.gov/), and the air pollutant data were obtained from the China Air Quality Online Monitoring and Analysis Platform Historical Data (https://www.aqistudy.cn/). The meteorological dataset encompasses daily mean temperature (°C), atmospheric pressure (kPa), relative humidity (%), and wind speed (m/s). Meanwhile, the air pollutant dataset includes daily concentration data for $PM_{2.5}$, $PM_{10}$, $NO_2$, $SO_2$, and the average concentration of $O_3$ over 8 consecutive hours. These data, sourced from official platforms, exhibit authority, authenticity, and timeliness.

## Statistical analysis

Statistical descriptions employed mean ± standard deviation, maximum, minimum, and quartile to characterize the data. Spearman's correlation coefficients were computed to evaluate the correlation among meteorological factors, air pollutants, and the daily incidence of HFMD. To mitigate collinearity, variables with correlation coefficients exceeding 0.8 were not simultaneously included in the model construction (*Du et al., 2019*; *Cheng et al., 2018*). The final model incorporated mean temperature, wind speed, $PM_{2.5}$, $SO_2$, and $O_3$.

Prior investigations have indicated that the impacts of environmental factors on infectious diseases extend beyond the day of exposure, with effects observed over several days or even weeks, often characterized by non-linear exposure-response relationships (*Hao et al., 2020*; *Yin et al., 2019*). Given the scattered nature of HFMD incidence, approximately adhering to a Poisson distribution, this study opted for a quasi-Poisson distribution instead of a Poisson distribution (*Hoef & Boveng, 2007*). Employing a distributional lag nonlinear model based on a generalized additive model, with a quasi-Poisson distribution as the function family, we analyzed the influence and lagged effects of environmental factors on HFMD incidence (*Cheng et al., 2014*; *Gasparrini, 2011*). The established regression model is as follows:

$Y_t \sim Poisson\ (u_t)$

*Log $(u_t)$ = α + cb (M, lag) + ns (Xi, df) + ns (Time, df) + DOW*

t denotes the observation date; Yt denotes the number of HFMD incidence on day t, and α denotes the intercept; cb denotes the cross-basis function used to assess the nonlinear relationship and lagged effects among meteorological factors, atmospheric pollutants, and the number of HFMD incidence; ns denotes the natural spline function; M denotes the study variables; Xi denotes the other environmental variables included in the model; Time denotes the seasonal trend and the long-term trend; and DOW to control the days of the week effect, and the test level α = 0.05. The degrees of freedom (*df*) of time, meteorological factors and air pollutants are determined by quasi-Poisson's Akaike Information Criteria (Q-AIC). The *df* of the time variable is six. Degrees of freedom for variables were as follows: mean temperature and wind speed (*df* = 5), $SO_2$ (*df* = 4), $PM_{2.5}$ and $O_3$ (*df* = 6). In order to avoid the existence of multicollinearity for air temperature and pressure (*r* = −0.906), PM2.5 and PM10 (*r* = 0.854), only air temperature, wind speed, $PM_{2.5}$, $SO_2$, and $O_3$ were included in the present modeling and study. Sensitivity analyses were conducted to control for seasonal and long-term trends by varying the time of year for df (6–8), and by varying df (4–6) for environmental factors.

Since the incubation period of HFMD is mostly 2–10 days and based on previous studies, lag-day pattern was finally set to 14 days in the analysis (*Hao et al., 2020*; *Gu, Li & Lu, 2020*). In the analysis of the single-day effect, the lag periodswere divided into six types (lag 0 days, lag 3 days, lag 7 days, lag 10 days, and lag 14 days). In the present study,We chose the 90th or 10th percentile as the cut-off point for extreme weather (*Zhang et al., 2019*) and the 90th percentile as the cut-off point for high air pollutant concentrations. Extreme meteorological factors ($P_5$th, $P_{10}$th, $P_{90}$th, $P_{95}$th) and high concentrations of air pollutants ($P_{90}$th, $P_{95}$th) at different percentiles were compared with their medians to obtain the relative risks (*RRs*) and 95% confidence intervals (*95% CIs*). R software (4.3.0) was used for all analysis. Significance tests were two-sided, and *P* < 0.05 was considered statistically significant.

## RESULTS

As shown in Table 1. The average number of HFMD cases per day was 12; the daily mean temperature, atmospheric pressure, humidity, and wind speed were 14.82 °C, 100.84 kPa, 67.30% and 2.29 m/s; the average concentrations of $PM_{2.5}$, $PM_{10}$, $NO_2$, $SO_2$, and $O_3$ were 50.77, 88.57, 32.62, 15.78 and 112.26 ug/m$^3$.

As depicted in Fig. 1, the daily incidence of HFMD exhibits a distinct seasonal pattern, characterized by a prominent peak from May to August and a minor peak in October to November each year. There has been a declining trend observed annually since 2018. Both meteorological factors and air pollutants demonstrate evident seasonality and cyclical patterns.

Table 2 presents the correlation coefficents among different variables. The correlation between temperature and HFMD stands out as the most substantial among meteorological factors (*r* = 0.457). Among air pollutants, $O_3$ exhibits the highest correlation coefficient with HFMD (*r* = 0.248).

Figure 2 visualizes the exposure-lag-response relationships of meteorological factors and air pollutants with HFMD using three-dimensional plots. The cumulative effects of meteorological factors and air pollutants on the incidence of HFMD are shown in the Fig. 3. A nonlinear associations are found between temperature, wind speed, $PM_{2.5}$, $SO_2$, $O_3$, and the risk of HFMD. The effect of temperature on the risk of HFMD development was generally upward, with the $RR$ reaching its maximum at about 34 °C ($RR$ = 2.411, 95% CI [1.251–4.646]); the relationship of wind speed with HFMD development was a "J" shape, with the RR reaching its maximum at about 6.5 m/s ($RR$ = 3.535, 95% CI [0.556–22.480]);. The exposure-response curves of $PM_{2.5}$ and HFMD were similar to a "W" shape, with the peak value occurring at a $PM_{2.5}$ concentration of 116 ug/m$^3$ ($RR$ = 1.921, 95% CI [1.131–3.262]). In addition, the risk of HFMD also showed an increasing trend with increasing $SO_2$ concentration, whereas an overall decreasing trend was presented with increasing $O_3$ concentration.

Figure 2 illustrates the exposure-lag-response relationships between meteorological factors, air pollutants, and HFMD through three-dimensional plots. The cumulative effects of these variables on HFMD incidence are depicted in Fig. 3. Nonlinear associations are evident between temperature, wind speed, $PM_{2.5}$, $SO_2$, $O_3$, and the risk of HFMD. The impact of temperature on HFMD risk displays an upward trend, reaching its peak at approximately 34 °C ($RR$ = 2.411, 95% CI [1.251–4.646]). The relationship between wind speed and HFMD follows a "J" shape, with the RR peaking at around 6.5 m/s ($RR$ = 3.535, 95% CI [0.556–22.480]). The exposure-response curves for $PM_{2.5}$ and HFMD exhibit a "W" shape, with the peak value occurring at a $PM_{2.5}$ concentration of 116 μg/m³ ($RR$ = 1.921, 95% CI [1.131–3.262]). Furthermore, the risk of HFMD demonstrates an increasing trend with rising $SO_2$ concentration, while an overall decreasing trend is observed with increasing $O_3$ concentration.

The impact of extreme low temperatures peaked on lag day 1 and subsequently demonstrated a decreasing trend, reaching a minimum at lag day 4. Extreme high temperatures reached their maximum effect on lag day 0 and exhibited a minimum at lag day 1. The influence of extreme low temperature on HFMD manifested more rapidly than the impact of extreme high temperature, with a higher risk observed at the $P_5$th compared to the $P_{10}$th. Both extreme high and low wind speeds emerged as significant risk factors for HFMD. For high wind speeds, the critical period was between lag days 9–14 of the $P_{95}$th and lag days 11–14 of the $P_{90}$th, with the risk of development showing an increasing trend with the exacerbation of extreme wind speed. Detailed information can be found in Fig. 4.

High $PM_{2.5}$ concentrations exhibited a risk effect on the incidence of HFMD on days 1 and 2 of the lag, with a higher risk observed at the $P_{95}$th compared to the $P_{90}$th. High SO2 concentrations were identified as a primary risk factor for HFMD incidence, showing statistically significant effects at lag 1 day and lag 9–12 days. The risk of single-day incidence was higher at the $P_{95}$th SO2 concentration compared to the $P_{90}$th. Refer to Fig. 5 for a detailed representation.

Table 3 displays the $RR$ for extreme high temperature and low wind speed at both the $P_{90}$th and $P_{95}$th. The $P_{95}$th $RR$ values for high temperature and low wind speed were observed to be higher than the corresponding $P_{90}$th $RR$ values. Cumulative effects of high

**Table 1 Description of meteorological factors, air pollutants and HFMD cases.**

| Variable | Mean ± Standard deviation | Percentile | | | | | | | | |
|---|---|---|---|---|---|---|---|---|---|---|
| | | Min | $P_5$ | $P_{10}$ | $P_{25}$ | $P_{50}$ | $P_{75}$ | $P_{90}$ | $P_{95}$ | Max |
| Case | 11.82 ± 19.85 | 0 | 0 | 0 | 1 | 4 | 12 | 35 | 57 | 146 |
| Temperature (°C) | 14.822 ± 10.51 | −12.24 | −1.30 | 0.30 | 4.90 | 15.02 | 24.48 | 28.00 | 29.40 | 34.44 |
| Atmospheric pressure (kPa) | 100.84 ± 0.99 | 98.41 | 99.35 | 99.53 | 99.94 | 100.87 | 101.64 | 102.14 | 102.40 | 103.45 |
| Relative humidity (%) | 67.30 ± 13.52 | 25.94 | 44.60 | 48.70 | 56.88 | 68.00 | 78.06 | 84.80 | 87.90 | 95.75 |
| Wind speed (m/s) | 2.29 ± 0.94 | 0.56 | 1.05 | 1.21 | 1.55 | 2.16 | 2.84 | 3.57 | 4.05 | 6.78 |
| $PM_{2.5}$ (ug/m$^3$) | 50.77 ± 34.01 | 3.00 | 15.00 | 20.00 | 28.00 | 41.00 | 63.00 | 96.00 | 123.00 | 241.00 |
| $PM_{10}$ (ug/m$^3$) | 88.57 ± 46.64 | 9.00 | 33.00 | 41.00 | 56.00 | 79.00 | 111.00 | 148.00 | 178.40 | 370.00 |
| $NO_2$ (ug/m$^3$) | 32.62 ± 16.56 | 5.00 | 12.00 | 14.00 | 20.00 | 29.00 | 43.00 | 57.00 | 64.00 | 114.00 |
| $SO_2$ (ug/m$^3$) | 15.78 ± 9.25 | 3.00 | 6.00 | 7.00 | 10.00 | 13.00 | 19.00 | 26.00 | 33.00 | 81.00 |
| $O_3$ (ug/m$^3$) | 112.26 ± 53.31 | 5.00 | 38.00 | 48.00 | 68.00 | 106.00 | 151.00 | 187.00 | 207.00 | 278.00 |

temperature were maximized at lags of 0–10 days for both the $P_{90}$th and $P_{95}$th, with *RR* values of 1.787 (1.244–2.569) and 1.880 (1.261–2.804), respectively. Similarly, cumulative lagged effects of low wind speed increased with the number of lag days. The cumulative effect of low wind speed $P_{90}$th and $P_{95}$th was largest at 14 day lag with *RR* values of 1.498 (1.283–1.750) and 1.702 (1.389–2.085), respectively. Additionally, the cumulative hazardous effects of extreme high winds ($P_{95}$th) were statistically significant at 14 day lag. For high $PM_{2.5}$ concentration, statistically significant hazard effects on HFMD were observed from lag 0–3 days to lag 0–14 days. The maximum effect was observed at the $P_{90}$th at 14 day lag, with an *RR* value of 1.637 (1.069–2.506), and at the $P_{95}$th at lag 0–10 days, with an *RR* value of 1.569 (1.021–2.411). Furthermore, higher concentration of $SO_2$ ($P_{95}$th) exhibited a statistically significant *RR* value at 14 day lag, with an *RR* value of 1.425 (1.001–2.030).

# DISCUSSION

Our investigation revealed a significant annual peak in the incidence of HFMD in Jining City, with a pronounced surge observed from May to August and a secondary, peak occurring between October and November. Nonlinear associations were identified between HFMD and environmental variables including temperature, wind speed, $PM_{2.5}$, $SO_2$, and $O_3$.

Time series analysis revealed an approximately linear association between temperature and the incidence of HFMD, where the risk of incidence exhibited a positive correlation with temperature escalation. Consistent with prior research in Guangzhou and Japan, a linear relationship between temperature and HFMD was reported (*Chen et al., 2014*; *Onozuka & Hashizume, 2011*). However, recent investigations have indicated an inverted "V" or "J" shape in the temperature-HFMD relationship (*Du et al., 2019*; *Dong et al., 2016*; *Yu et al., 2019*). These divergent trends may be attributed to variations in climate characteristics and regional indicators across different geographical locations, encompassing factors such as healthcare infrastructure and economic conditions.

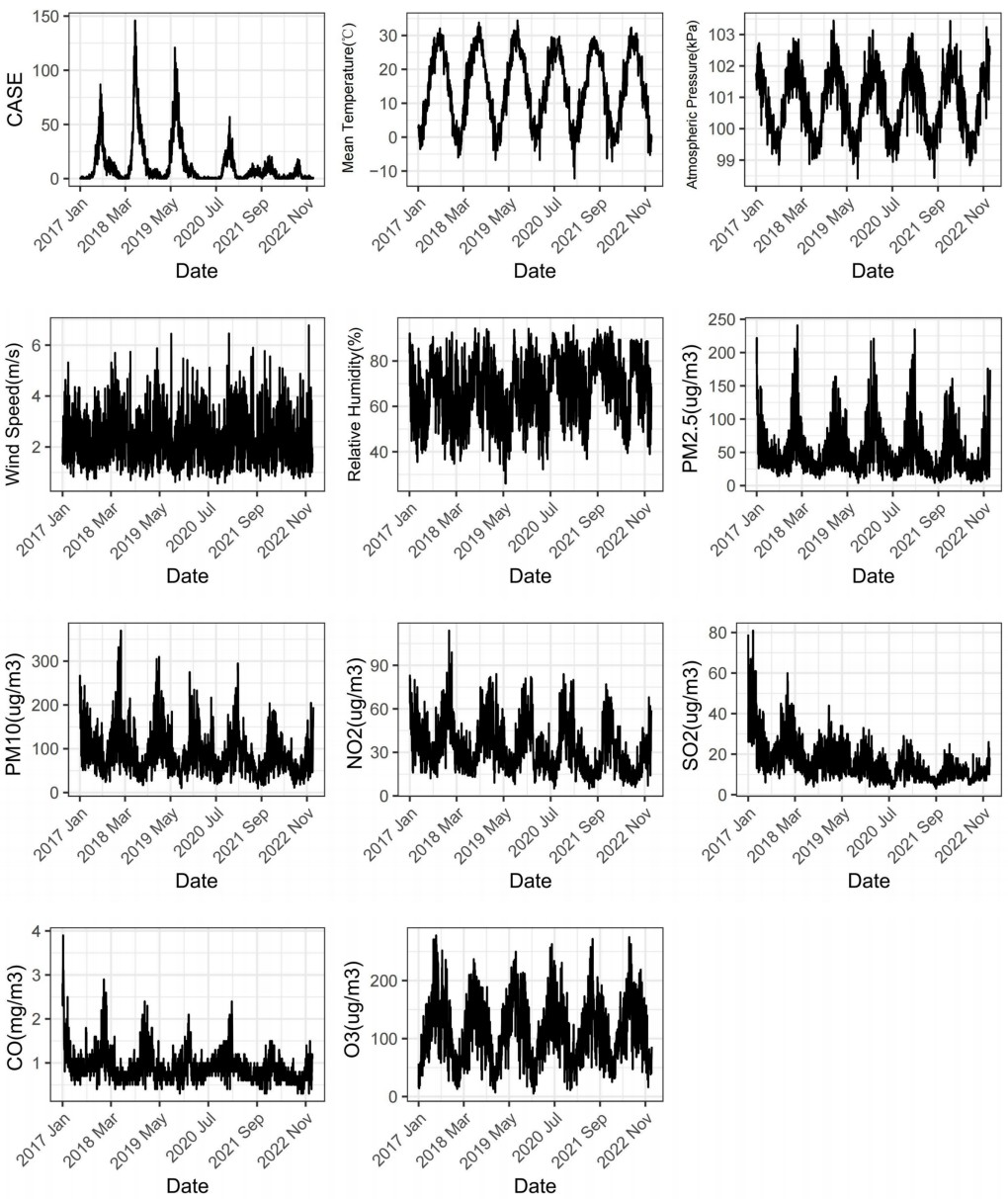

**Figure 1 Changes of meteorological factors, air pollutants and the number of HFMD cases during 2017–2022.**

Furthermore, discrepancies in model design, including lag days and covariables, also contribute to the observed differences (*Yu et al., 2018*). High and low temperature exhibit contrasting impact on HFMD. The deleterious effects of cold manifest more rapidly than those of hot, potentially attributable to a deterioration in personal hygiene behaviors during the winter season, thereby elevating the risk of infection. Additionally, the decrease in children's immunity during winter accelerates the occurrence of the disease (*Chiu et al., 2020*; *Lin et al., 2014*). The typical incubation period for HFMD spans 3–7 days, during which elevated temperatures contribute to an increased disease incidence, with a higher risk observed at the $P_{95}$th compared to the $P_{90}$th. Within an optimal temperature range,

**Table 2 Correlation analysis of meteorological factors and air pollutants with the number of HFMD cases in Jining City.**

| Variable | Case | Temperature | Atmospheric pressure | Relative humidity | Wind speed | $PM_{2.5}$ | $PM_{10}$ | $NO_2$ | $SO_2$ | $O_3$ |
|---|---|---|---|---|---|---|---|---|---|---|
| Case | 1 | | | | | | | | | |
| Temperature | 0.457* | 1 | | | | | | | | |
| Atmospheric pressure | −0.337* | −0.906* | 1 | | | | | | | |
| Relative humidity | 0.015 | 0.097* | −0.140* | 1 | | | | | | |
| Wind speed | −0.118* | −0.002 | −0.118* | −0.128* | 1 | | | | | |
| $PM_{2.5}$ | −0.194* | −0.534* | 0.485* | 0.01 | −0.214* | 1 | | | | |
| $PM_{10}$ | −0.148* | −0.460* | 0.405* | −0.223* | −0.085* | 0.854* | 1 | | | |
| $NO_2$ | −0.02 | −0.569* | 0.599* | −0.150* | −0.347* | 0.663* | 0.684* | 1 | | |
| $SO_2$ | 0.060* | −0.305* | 0.309* | −0.422* | −0.108* | 0.496* | 0.620* | 0.618* | 1 | |
| $O_3$ | 0.248* | 0.740* | −0.661* | −0.218* | −0.022 | −0.275* | −0.179* | −0.434* | −0.058* | 1 |

**Note:**
*$P < 0.05$.

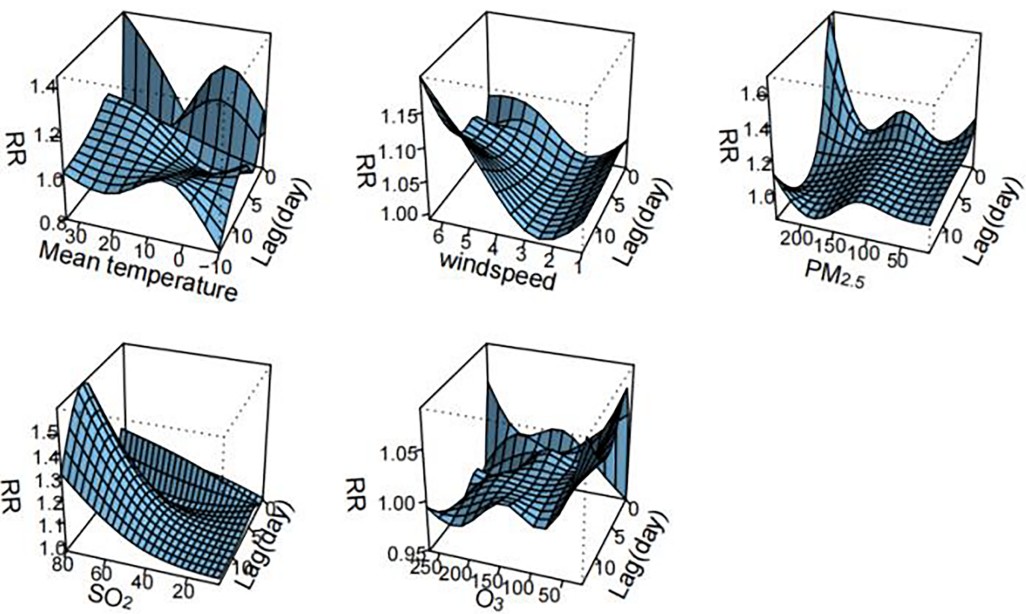

**Figure 2 Three-dimensional delayed response curve of meteorological factors and air pollutants to the incidence of HFMD ($n = 14$).**

high temperatures facilitate the replication and transmission of enterovirus (*Fong & Lipp, 2005*; *Rajtar et al., 2008*). The cumulative effect of high temperature on HFMD peaked at 14 days, which is consistent with the findings of Guilin (*Yu et al., 2019*). In contrast, Shijiazhuang reported a peak cumulative effect after 10 days, possibly attributed to lower temperatures in Shijiazhuang compared to Jining city, resulting in a relatively shorter duration of disease impact on the population (*Liu et al., 2022*).

Our investigation revealed a 'J'-shaped dose-response relationship between wind speed and HFMD, indicating that both low and high wind speeds contribute to the promotion of HFMD occurrence. Consistent findings from studies in Malaysia and the Yili region support the significant impact of wind speed on HFMD incidence (*Wahid, Suhaila &*

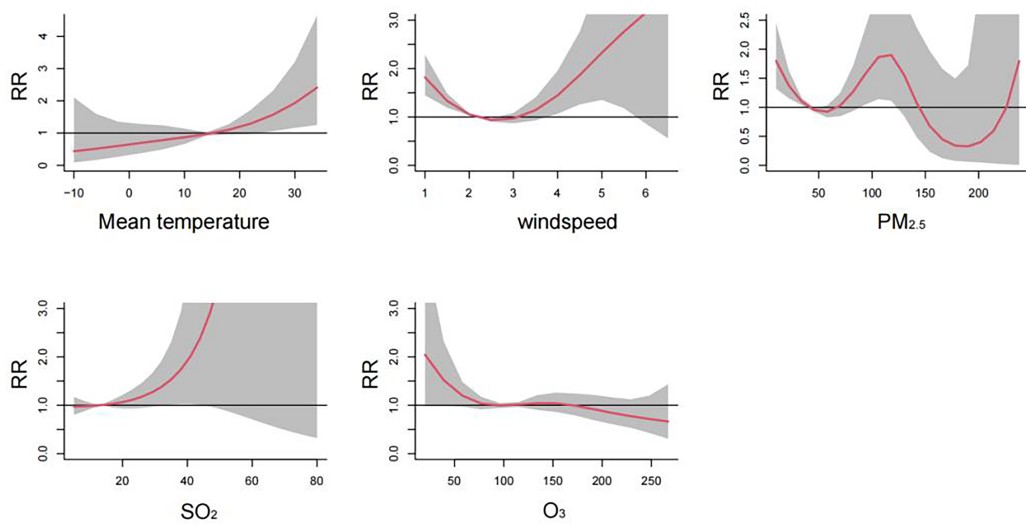

**Figure 3 Cumulative effects of meteorological factors and air pollutants on the incidence of HFMD (*n* = 14).**

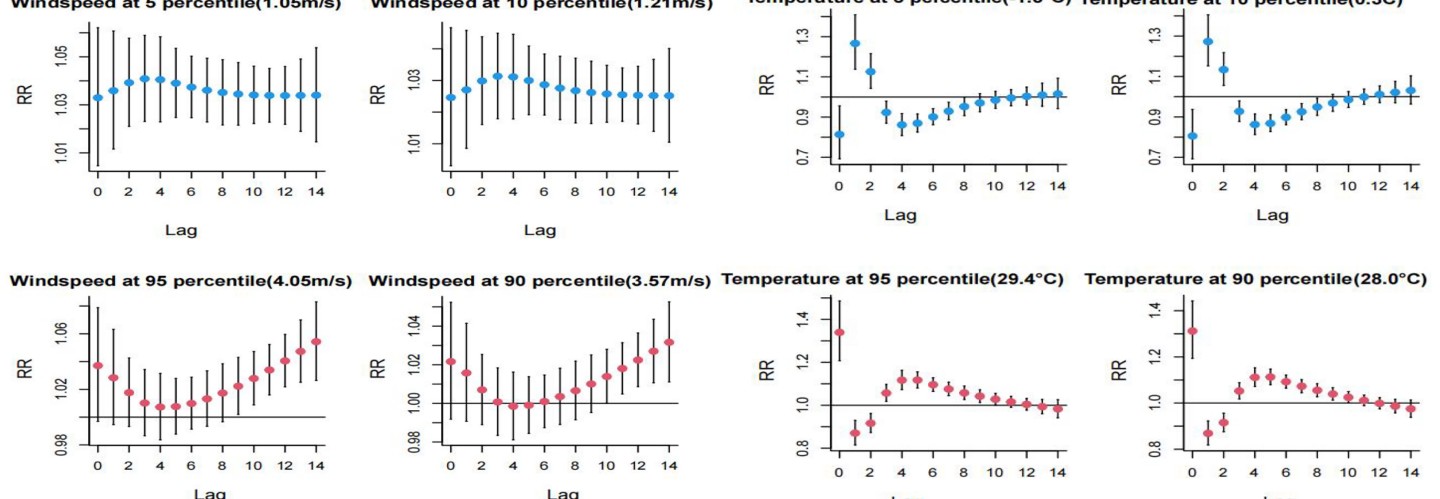

**Figure 4 Distributional lag effects of extreme meteorological factors on the incidence of hand, foot and mouth disease in different lag days.** Select the middle value of each variable as a reference.

*Rahman, 2021*; *Yi et al., 2021*). Notably, we observed that the $P_{95}$th wind speed posed a greater risk than the $P_{90}$th. Parallel research in Hefei City indicated that low wind speed correlates with increased HFMD incidence (*Zhang et al., 2019*). This phenomenon may be attributed to heightened child activity during periods of low wind speed, leading to increased exposure to the HFMD virus. Our study demonstrated that the cumulative effect of high wind speed was statistically significant only on days 14 at the $P_{95}$th. While high wind speed facilitated the spread of the HFMD virus, the rapid dilution of virus content in droplets by high wind speed resulted in a hysteresis effect, diminishing the danger after the initial period (*Liu et al., 2022*). Contrary findings have been reported, as demonstrated by

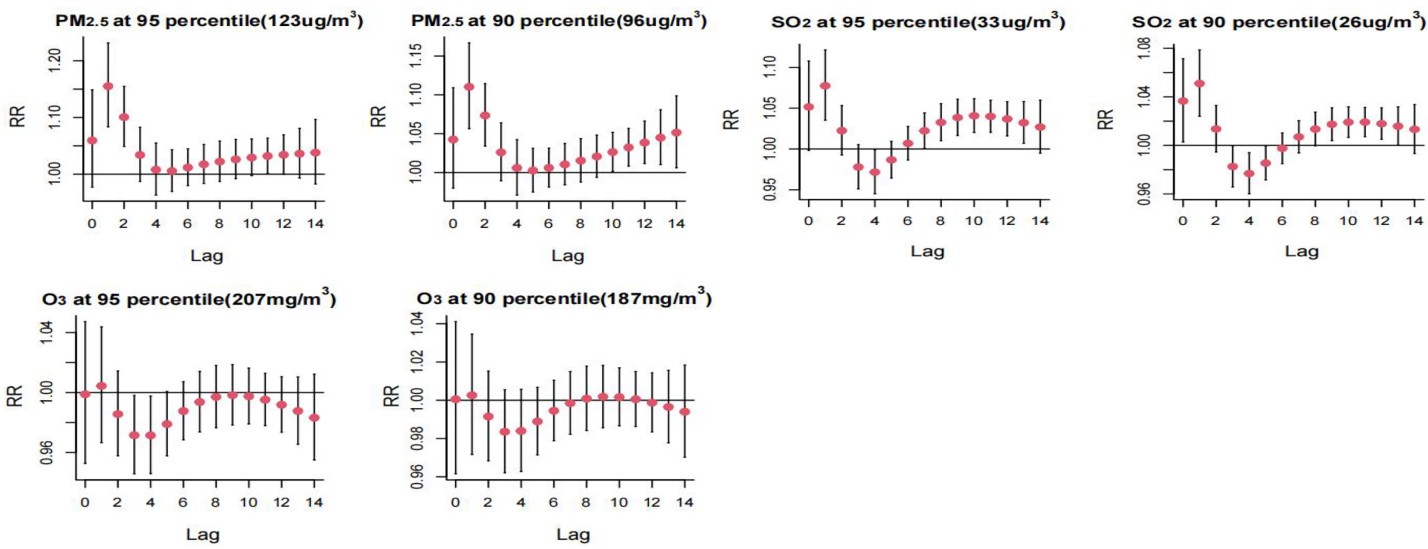

**Figure 5 Distributed lag effect of high concentration air pollution on the incidence of hand, foot and mouth under different lag days. Select the middle value of each variable as a reference.**

**Table 3 Cumulative effects of extreme meteorological factors and high concentrations of air pollutants on the incidence of HFMD.**

| Variable | Relative risk [95% CI] | | | | | |
|---|---|---|---|---|---|---|
| | Lag0 | Lag0-3 | Lag0-5 | Lag0-7 | Lag0-10 | Lag0-14 |
| Temperature | | | | | | |
| P5 | 0.816 [0.635–1.05] | 1.067 [0.778–1.464] | 0.800 [0.533–1.200] | 0.671 [0.414–1.089] | 0.611 [0.334–1.120] | 0.618 [0.286–1.333] |
| P10 | 0.806 [0.640–1.015] | 1.078 [0.810–1.436] | 0.807 [0.558–1.167] | 0.671 [0.432–1.043] | 0.608 [0.350–1.058] | 0.647 [0.321–1.305] |
| P90 | 1.312 [1.135–1.517] | 1.098 [0.913–1.320] | 1.357 [1.066–1.729] | 1.591 [1.187–2.132] | 1.787 [1.244–2.569] | 1.736 [1.113–2.708] |
| P95 | 1.339 [1.143–1.570] | 1.128 [0.920–1.383] | 1.408 [1.078–1.839] | 1.669 [1.201–2.292] | 1.880 [1.261–2.804] | 1.871 [1.149–3.047] |
| Windspeed | | | | | | |
| P5 | 1.033 [0.990–1.078] | 1.157 [1.061–1.263] | 1.251 [1.117–1.402] | 1.345 [1.175–1.539] | 1.489 [1.262–1.758] | 1.702 [1.389–2.085] |
| P10 | 1.025 [0.992–1.058] | 1.118 [1.046–1.194] | 1.187 [1.089–1.294] | 1.255 [1.132–1.390] | 1.356 [1.195–1.539] | 1.498 [1.283–1.750] |
| P90 | 1.022 [0.976–1.069] | 1.046 [0.957–1.143] | 1.043 [0.931–1.170] | 1.048 [0.913–1.203] | 1.080 [0.904–1.291] | 1.192 [0.950–1.495] |
| P95 | 1.037 [0.976–1.102] | 1.096 [0.973–1.235] | 1.113 [0.945–1.298] | 1.138 [0.945–1.372] | 1.217 [0.957–1.547] | 1.446 [1.068–1.957] |
| PM2.5 | | | | | | |
| P90 | 1.042 [0.948–1.146] | 1.275 [1.064–1.527] | 1.286 [1.019–1.622] | 1.306 [1.148–1.487] | 1.39 [1.002–1.928] | 1.637 [1.069–2.506] |
| P95 | 1.059 [0.936–1.199] | 1.393 [1.107–1.753] | 1.412 [1.047–1.904] | 1.307 [1.000–1.708] | 1.569 [1.021–2.411] | 1.224 [1.034–3.136] |
| SO2 | | | | | | |
| P90 | 1.037 [0.985–1.090] | 1.085 [0.991–1.188] | 1.044 [0.929–1.174] | 1.049 [0.912–1.207] | 1.247 [0.941–1.653] | 1.177 [0.944–1.468] |
| P95 | 1.052 [0.971–1.139] | 1.133 [0.981–1.307] | 1.086 [0.9–1.311] | 1.118 [0.892–1.400] | 2.720 [0.744–9.935] | 1.425 [1.001–2.030] |
| O3 | | | | | | |
| P90 | 1.000 [0.942–1.063] | 0.978 [0.871–1.099] | 0.952 [0.819–1.107] | 0.945 [0.791–1.129] | 0.949 [0.765–1.179] | 0.940 [0.721–1.224] |
| P95 | 0.999 [0.929–1.074] | 0.961 [0.833–1.108] | 0.914 [0.761–1.098] | 0.897 [0.722–1.114] | 0.891 [0.684–1.160] | 0.854 [0.621–1.173] |

the Sichuan study, which wind speed showed no positive correlation with HFMD (*Song et al., 2019*). In a Guilin study, both high and low wind speeds were found to have a protective effect on HFMD (*Yu et al., 2019*). Conversely, a study in Guangzhou found no correlation between wind speed and HFMD (*Huang et al., 2013*). The observed disparities across studies may be attributed to geographical, social, and economic variations among different regions.

Limited research has explored the associations between particulate matter (PM) and HFMD, yielding varied results. Yin's investigation identified $PM_{10}$ as a risk factor for HFMD, in contrast to He's study, which reported a negative correlation between $PM_{10}$ concentration and HFMD (*Yin et al., 2019*; *He et al., 2020*). However, the Ningbo study found no correlation between $PM_{10}$ and HFMD (*Huang et al., 2016*). Our study results reveal that elevated $PM_{2.5}$ concentration poses a risk for HFMD. Specifically, at the $P_{90}$th PM2.5 concentration, the cumulative risk increases with lag days, reaching a maximum at a 14-day lag. However, at the $P_{95}$th pollutant concentration, the cumulative risk attained its maximum after a 10-day lag, demonstrating a subsequent decreasing trend. This observed trend may be attributed to exceptionally high pollutant concentrations, prompting corresponding governmental measures to safeguard public health, such as school closures and advisories for residents to stay indoors. Concurrently, individuals adopt self-protective measures, including increased water consumption to maintain respiratory tract moisture and the use of masks during going out, thereby reducing pollutant-related risks (*Allen & Barn, 2020*; *Onishi, 2017*). Peng's study also identified high concentrations of both $PM_{10}$ and $PM_{2.5}$ as HFMD risk factors, suggesting that children may come into contact with HFMD viruses adhered to particulate matter, facilitating transmission through the fecal-oral route or close contact (*Peng et al., 2022*). Moreover, considering the respiratory tract as one of the transmission routes for HFMD, exposure to fine particulate matter heightens the body's inflammatory response and exacerbates reactions to respiratory viral infections (*Smith et al., 2006*). The extensive surface area of PM2.5 and PM10 provides attachment sites for HFMD viruses, allowing them to enter the lungs and bloodstream through the respiratory system (*Xiao et al., 2017*; *Lee et al., 2014*). Additionally, airborne particulate matter may obstruct sunlight's ultraviolet light, with sterilizing properties, thereby prolonging the survival time of viruses in the air (*Huang et al., 2016*; *Shimasaki et al., 2016*).

The exposure-response relationship SO2 concentration and HFMD exhibited a 'J'-shaped pattern, aligning with findings in Ningbo (*Gu, Li & Lu, 2020*). The lagged effect of a single day at high concentrations was more often shown as a risk factor, and the $P_{95}$th was higher than the $P_{90}$th. A study in Shijiazhuang City reported no significant effect of high $SO_2$ concentration on HFMD (*Liu et al., 2022*). Conversely, elevated $SO_2$ concentrations were associated with increased HFMD incidence in a study in Hefei City, consistent with our findings (*Wei et al., 2019*). $SO_2$ is a major air pollutant primarily stemming from fossil fuel combustion in daily life and chemical industry activities (*Lou et al., 2017*), which has been linked to adverse health effects in children, including increased disease incidence such as eczema in studies conducted in the United States (*Kathuria & Silverberg, 2016*). Moreover, research suggests that $SO_2$ can impact the normal flora of the intestinal tract,

potentially affecting children's immunity (*Beamish, Osornio-Vargas & Wine, 2011*). Wigenstam's study further revealed that $SO_2$, beyond its toxic impact on the respiratory system, serves as a risk factor for inflammation development, thereby heightening susceptibility to diseases in children (*Wigenstam et al., 2016*).

The dose-response relationship between $O_3$ and HFMD was observed to exhibit an 'S'-shaped pattern, consistent with findings from a study conducted in Shijiazhuang (*Liu et al., 2022*). High concentrations of $O_3$ demonstrated no impact on the single-day onset of HFMD at the $P_{90}$th. However, it exerted a protective effect on the lagged 3rd and 4th days at the $P_{95}$th. Research conducted in Guilin similarly identified that elevated concentrations of $O_3$ conferred a protective effect against HFMD (*Yu et al., 2019*). However, a study in Ningbo found that $O_3$ was a risk factor for HFMD, which may be explained by the fact that the effect of temperature on HFMD was not controlled for in the Ningbo analysis (*Huang et al., 2016*). Furthermore, the higher temperature in Ningbo compared to Jining concurrently suggests that the effect of ozone on humans intensifies at elevated temperatures, as supported by previous studies (*Li et al., 2022*; *Shi et al., 2020*). $O_3$ exhibits not only bactericidal and antiviral characteristics but also facilitates wound healing, positioning $O_3$ therapy as a potential method for tissue repair (*Lim et al., 2019*). Animal experiments have revealed that exposure to ozone prolongs cell survival and stimulates the production of pertinent cytokines at optimal ozone concentrations, thereby restricting viral spread by inhibiting their propagation (*Ching, Juan & Cheng, 2007*).

Our research has several limitations. Firstly, our examination focused solely on HFMD incidence data in Jining City, potentially restricting the generalizability of our findings. Secondly, the study could not comprehensively account for all potential confounding factors, including geographic influences, host susceptibility, and preventive measures. Thirdly, the data collection relied on passive testing, possibly leading to an underestimation of the true number of HFMD cases. Lastly, our analysis primarily explored the impact of a single environmental factor variable on HFMD incidence. Future investigations should consider the effects of multiple environmental variables on infectious disease incidence, employing approaches such as the analysis of HFMD incidence through bioindices.

## CONCLUSIONS

In conclusion, there were significantly lagged effects of temperature, wind speed, $PM_{2.5}$, $SO_2$ and $O_3$ on HFMD incidence. High temperature, low wind speed, high concentrations of $PM_{2.5}$ and $SO_2$ were associated with increased risk for HFMD. This study contributes to the understanding of the impacts of extreme meteorological conditions and elevated levels of air pollutants on HFMD, which provided a scientific basis for relevant authorities to implement targeted preventive measures under the contexts of extreme meteorological events and high air pollution levels. These findings have great significance for the establishment and improvement of an early warning system for HFMD, facilitating proactive interventions in public health.

## ACKNOWLEDGEMENTS

We thank the Jining Center for Disease Control and Prevention for providing the data on notified hand, foot and mouth disease cases.

### Funding

This work was supported by the National Key Projects of Research and Development of China (No. 2016YFC0900605). The funders had no role in study design, data collection and analysis, decision to publish, or preparation of the manuscript.

### Grant Disclosures

The following grant information was disclosed by the authors:
National Key Projects of Research and Development of China: 2016YFC0900605.

### Competing Interests

The authors declare that they have no competing interests.

### Author Contributions

- Haoyue Cao conceived and designed the experiments, performed the experiments, analyzed the data, prepared figures and/or tables, authored or reviewed drafts of the article, and approved the final draft.
- Rongrong Xu conceived and designed the experiments, performed the experiments, analyzed the data, prepared figures and/or tables, authored or reviewed drafts of the article, and approved the final draft.
- Yongmei Liang analyzed the data, prepared figures and/or tables, and approved the final draft.
- Qinglin Li analyzed the data, authored or reviewed drafts of the article, and approved the final draft.
- Wenguo Jiang analyzed the data, prepared figures and/or tables, and approved the final draft.
- Yudi Jin analyzed the data, authored or reviewed drafts of the article, and approved the final draft.
- Wenjun Wang conceived and designed the experiments, performed the experiments, analyzed the data, authored or reviewed drafts of the article, and approved the final draft.
- Juxiang Yuan conceived and designed the experiments, performed the experiments, analyzed the data, authored or reviewed drafts of the article, and approved the final draft.

### Data Availability

The code and raw data are available in the Supplemental Files.

### Supplemental Information

Supplemental information for this article can be found online at http://dx.doi.org/10.7717/peerj.17163#supplemental-information.

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
