# Peer review of "Effects of extreme meteorological factors and high air pollutant concentrations on the incidence of hand, foot and mouth disease in Jining, China"

_PeerJ, doi:10.7717/peerj.17163_

## Round 0.1 · original submission · Major Revisions

The reviewers generally commend the manuscript's scientific validity but identify areas for improvement. Reviewer 1 highlights language issues and recommends involving an English-proficient colleague. They also suggest adding labels to Figure 1. Reviewer 2 suggests structural enhancements, emphasizing the need for a climate description in the study area and considering air humidity's impact on HFMD. Reviewer 3 notes insufficient support for the article's novelty and raises concerns about unobserved confoundings in the experimental design. All reviewers call for a discussion of limitations. Addressing language, structural, and content concerns will strengthen the manuscript's clarity and scientific rigor.

**Language Note:** The Academic Editor has identified that the English language must be improved. PeerJ can provide language editing services - please contact us at copyediting@peerj.com for pricing (be sure to provide your manuscript number and title). Alternatively, you should make your own arrangements to improve the language quality and provide details in your response letter. – PeerJ Staff

Reviewer 1 ·

Basic reporting

Overall, the manuscript is written in clear, and unambiguous language, but minor grammar errors were detected. Sufficient references and background are provided. Figures and tables included are relevant to the hypothesis. The English language should be improved to ensure that an international audience can clearly understand your text. Some examples where the language could be improved:
- In line 116, “the” is not needed in “to avoid the collinearity”, and “when the correlation coefficients between two variables were larger than” instead of “large than”.
- In line 117, “the two variables were not included in the same model.” “the final included variables” can be changed to “the variables included in the final model.”
- In line 155, “the average number of HFMD cases per day was 12 cases” should be changed to “the average number of HFMD cases per day was 12.”
I suggest you have a colleague who is proficient in English and familiar with the subject matter review your manuscript or contact a professional editing service.

In addition, for Figure 1, please add labels to the X and Y axes.

Experimental design

- Lines 136-139 are a bit hard to understand. How do you choose the degrees of freedom for a variable? Does it mean you tried classifying a variable into different numbers of groups by quantiles? And what models were the AIC derived from? Were those univariate or multivariate models? Were those lag nonlinear models combined with Poisson generalized additive regression?

- You could rephrase lines 136-139 to something like “Continuous variables were further classified into quantiles. AIC was used to assess the model fit, with a smaller value indicating better model fit.” And please provide more details on what models were used.

Validity of the findings

- What are some of the limitations of your study? Could you describe those in your discussion section?

- Have you done any power calculations? Does your study have a sufficient sample size? In Figure 3, could you explain why there are large confidence intervals at the right or left end of the x-axis? How valid are those values at the right or left end of the x-axis? In Figure 4, why are the confidence intervals from the windspeed model much higher than those from the temperature model?

Reviewer 2 ·

Basic reporting

Scientifically valid manuscript, with a sufficient backgroud. Structure should be improved, and some other minor comments below.

Experimental design

Research questions well defined, methods are described with suffiicient details.

Study Area section should be supplemented by a description of climate (with references).

Air humidity should be identified as an important factor to study its effect on the incidence of HFMD

Validity of the findings

No comments

Additional comments

Abstract:
1) in Results your show the cumulative effect of extreme high wind speed (R48), but controversial Conclusion shows low wind speed as a risk factor (R53). Please correct;
2) in Results, it is not clear what P90 and P95 mean, please correct.

R94: Please add a description of the climate in the study area.

R107 (R155 and elsewhere below): more correct is a term daily ‘mean’ (not average) temperature

R118: why was relative humidity not included in the model? Given that the results of previous studies show that air humidity is a very important factor for the spread of HFMD, I would say that this is a critical limitation of the current study.

R165 and R178: These conclusions on the correlation with O3 seem to be contradictory, please explain.

R256-261: please allocate these sentences into a separate paragraph, and strengthen the importance of the preventive measures (with references).

Please structure the Discussion section into several separate paragraphs.

In Discussion, please discuss the limitations of your current study, and plans for a future research (Definitely, special studies should be conducted on the combined effect of weather factors – air temperature, humidity and wind speed, let say, in the form of bioclimatic indices – on the incidence of HFMD).

Table 1: please add units to all variables. In you study you discuss the correlation of HFMD with extreme air temperature, wind speed, etc. Please add extreme values (5%-, 10%-, 90%- and 95%-percentiles) to the Table 1.

Annotated reviews are not available for download in order to protect the identity of reviewers who chose to remain anonymous.

Reviewer 3 ·

Basic reporting

Literature references, sufficient field background/context provided can not strongly support the novelty of the article.

Experimental design

The authors collected data on daily incidence cases of HFMD, meteorological factors and air pollution for Jining City from 2017 to 2022. Distributed lag nonlinear models combined with Poisson generalized additive regressionwere used to explore the effects of extreme meteorological factors and high concentrations of pollutants on HFMD. This is a general method, which still not deal with some unobserved confoundings.

Validity of the findings

Not enough in Sensitivity analysis.

---

## Round 0.2 · accepted · Accept

The original Academic Editor is not available so I have taken over handling this submission.

After a thorough review process, we have received positive feedback from all three reviewers regarding the quality and significance of your work. The reviewers were impressed with the novelty of your findings, the methodology employed, and the clarity of your presentation. They believe that your manuscript makes a valuable contribution to the field and merits publication in our journal.

Reviewer 1 ·

Basic reporting

The authors have adequately addressed my comments. Therefore, I have no further comments.

Experimental design

The authors have adequately addressed my comments. Therefore, I have no further comments.

Validity of the findings

The authors have adequately addressed my comments. Therefore, I have no further comments.

Reviewer 2 ·

Basic reporting

scientificly valid manuscropt, suitable to join the scholarly literature

Experimental design

correct

Validity of the findings

All underlying data have been provided; they are robust, statistically sound, & controlled

Additional comments

no comments

Reviewer 3 ·

Basic reporting

The revised version is much better.

Experimental design

There have been many improvements.

Validity of the findings

There have been many improvements.

Additional comments

I agree to publish it in its present form